

# Acoustic Sensors data transmission integrity and endurance with IoT-enabled location-aware framework

Shujaat Ali[1], Muhammad Nadeem[1], Sheeraz Ahmed[2], Faheem Khan[3], Murad Khan[4] and Abdullah Alharbi[5]

[1] Department of Computer Science, Faculty of Computing and Information Technology, International Islamic University, Islamabad, Punjab, Pakistan
[2] Department of Computer Science, IQRA National University, Peshawar, kpk, Pakistan
[3] Department of Computer Engineering, Gachon University, Seongnam, Seoul, Republic of South Korea
[4] Department of Computer Science and Engineering, Kuwait College of Science and Technology, Doha District, Kuwait
[5] Department of Computer Science, Community College, King Saud University, Riyadh, Saudi Arabia

Corresponding authors
Faheem Khan, faheem@gachon.ac.kr
Abdullah Alharbi, arharbi@ksu.edu.sa

## ABSTRACT

Environmental monitoring and disaster mitigation are critical applications of underwater acoustic sensor networks (UASNs). However, UASNs face significant challenges, including high latency, limited bandwidth, and energy constraints. This study introduces an Internet of Things (IoT)-driven location-aware framework (ILAF) designed to enhance UASN performance by utilizing non-GPS geographic coordinates for determining the location of sensor and sink nodes, identifying their neighbors based on coordinates and transmission range, and optimizing node placement and routing without the need for GPS modems. The framework is compared with several state-of-the-art protocols, including Bald Eagle Search inspired optimized energy efficient routing protocol (BES-OEERP) and IoT-enabled depth-based routing technique (IDBR), demonstrating superior performance. Specifically, ILAF achieved a packet delivery ratio (PDR) of 99%, which outperforms energy-efficient region-based source distributed routing algorithm (EERSDRA) (98%) and energy-efficient geo-opportunistic routing protocols (EEGORP) (96%). Additionally, ILAF reduced energy consumption by 20% compared to these existing protocols. These improvements result in a more energy-efficient network with fewer dead nodes (12 after 1,000 rounds) and higher throughput (5.7 kbps at 1,000 rounds), making ILAF suitable for real-time underwater applications. Future research will explore integrating lightweight IoT protocols like Message Queue Telemetry Transport (MQTT) and Constrained Application Protocol (CoAP) to enhance the framework's performance and reliability further.

# INTRODUCTION

Underwater acoustic sensor networks are becoming increasingly important because of their wide range of applications and significance. Sensor nodes consistently and regularly detect

environmental parameters as needed by an application. Monitoring and observing diverse underwater occurrences in vast aquatic areas are crucial for protecting the environment, facilitating oil and gas exploration, mitigating disasters, helping navigation, and conducting military surveillance (*Sun, Cui & Chen, 2021*). The acoustic sensor network consists of both mobile and stationary sensor nodes. Most sensor nodes alter their position in response to water waves. Autonomous underwater vehicles collect data from sensor nodes located in different areas. The buoys are placed on the seabed or within the water column. The sensor nodes deployed at various locations in the water are designed to collect data on multiple aspects, including salinity, temperature, depth, and different types of pollution. Data is transmitted to its destination with packets. Numerous intermediate and terminal nodes are located on boats or coastal stations. Collecting felt information in this way is unconventional. This system offers a cost-effective method of regularly collecting sensed information from numerous locations in underwater areas for extended durations (*González-García et al., 2020*).

There are several challenges associated with underwater sensor communication. Nevertheless, acoustic waves surpass electromagnetic waves and optical communication despite their limitations. The development of routing protocols for localization and routing protocols has been driven by the goal of improving the reliability and longevity of acoustic sensor networks (*Gola & Arya, 2023*).

Compared to radio frequency (RF) waves, underwater acoustic waves have longer propagation delays, a smaller bandwidth, and higher error rates in water depths. Acoustics works with frequencies between 1 and 100 kHz, and the time it takes to send can be between milliseconds and several seconds. The distance between the sending and receiving sensor nodes in space affects the delays. Since sound travels more slowly through water than air, it becomes much weaker because of solid absorption processes. Due to things like node movement and running into new obstacles, the underwater network world is always changing. This means that flexible network designs are needed. Different sound speeds on the ocean floor compared to those on the top of the water make things even more complicated. This can lead to a lot of multipath fading. The complicated interactions between ocean currents, marine life, and shipping activities can also cause waves. These interactions can mislead signals. The movement of the nodes can cause the Doppler effect, which can happen when the transmitter and receiver move closer or farther apart. All these factors combine with depth and absorption factors to cause intermittent connectivity and changes in link quality (*Mateen et al., 2019*).

Many research works have concentrated on Internet of Things (IoT) mechanisms (*Al-Atawi, Khan & Kim, 2022*; *Khan, Tarimer & Taekeun, 2022*; *Guo et al., 2023*; *Abbas et al., 2022*; *Khan et al., 2021*; *Al-Kahtani, Khan & Taekeun, 2022*; *Khan et al., 2022*). However, to tackle energy efficiency and dependable data delivery in underwater wireless sensor networks (UWSNs), IoT-enabled depth-based routing (IDBR) techniques consider depth information and counts on energy-efficient path selection. At the same time, Bald Eagle Search inspired optimized energy efficient routing protocol (BES-OEERP) applies the Bald Eagle Search algorithm for intelligent route and path selection, considering factors such as node depth, remaining energy, and signal strength. Both protocols show good

results in terms of improved energy efficiency, notable packet delivery ratio, and network lifetime when compared to baseline methods. underwater structure and arrangement of the ocean floor are crucial in the deployment and improvement of underwater acoustic sensors, anchor nodes, and sink nodes. Geographic information improves data transmission, tracking, and energy efficiency. Locating sensor nodes improves data sharing. By strategically assessing and choosing data packet paths, networks can choose the fastest channels. Communication is more reliable, and delays are reduced. Tracking the ship and monitoring the sea life requires proper placement of the sensor nodes. It is crucial to network energy efficiency. Find the neighbor and sensors based on their placements to join two nodes directly. This works best with nearby nodes. This can save lots of energy (*Kapileswar & Phani Kumar, 2022*).

Adding geographic information makes terrestrial wireless communication work much better. It is very important to know exactly where the nodes are in underwater wireless sensor networks to get around the problem of sound signal propagation. This then allows smart choices about forwarding data based on on-site routing algorithms. Less latency is achieved by carefully choosing the best sink nodes or sensor nodes for forwarding. This cuts down on the lengths that signals have to travel. This method improves network stability by reducing noise and signal interference, which means that data does not have to be sent again as often (*Farooq et al., 2021*). The energy and power consumption of the sensor nodes decreases significantly when the position of the nearest sink and sensor node for packet forwarding is known. Considering the limited energy resources of sensor nodes deployed in water, it is imperative to use energy-conserving approaches and solutions to prolong the operational lifespan of the network for ongoing monitoring duties. Therefore, a decentralized method of determining the location of nodes, together with a routing system that relies on the nodes' reported positions, can facilitate informed decision-making for forwarding data, enhance reliability, and optimize the long-term viability of the acoustic network (*Karim et al., 2021a*).

This study introduces the IoT-driven location-aware framework (ILAF), which enhances the reliability and longevity of underwater acoustic sensor networks (USANs) by utilizing non-GPS geographic coordinates for regions, sensor, and sink nodes, and neighbor identification with moving sink nodes' proximity to all sensor nodes and minimizing forwarder nodes. ILAF is designed to improve energy efficiency, reliability, and operational lifespan by strategically placing sink nodes, reducing communication distances, and balancing energy consumption.

## Research objectives and questions

The primary objective is to improve the energy efficiency, reliability, and network longevity of UASNs by introducing ILAF, a novel framework that utilizes non-GPS geographic coordinates for sensor node location identification; regions and sensor nodes that fall in the same regions are neighbors without computational power. This framework optimizes energy consumption and enhances data packet delivery without needing GPS modems. Additionally, ILAF seeks to minimize the distance between sink and sensor nodes to

improve reliability and minimize the need for relay nodes, decrease packet loss, and minimize delay.

- How does using non-GPS geographic coordinates impact the location of sensor nodes, region formation, and neighbor identification for energy efficiency and reliability of UASNs compared to existing location methods?
- How do moving sink nodes eliminate the need for forwarder nodes and improve packet loss by limiting the distance between sink and sensor nodes?
- Can the proposed ILAF framework enhance packet delivery and network longevity in highly dynamic underwater environments?
- What are the benefits and potential limitations of incorporating mobile sink nodes for efficient data collection in UASNs?

## Contributions

- Geographic coordinates utilization: Unlike traditional methods that depend on heuristic algorithms or specific metrics like depth, our approach uses non-GPS geographic coordinates to identify sensor nodes and their neighbors, including regions and virtual square areas. This technique ensures balanced energy consumption and higher reliability in packet delivery by accurately identifying sensor nodes without needing GPS modems.
- Strategic sink node placement: The framework enhances reliability and energy efficiency by optimally placing sink nodes based on geographic coordinates. This strategic placement minimizes the communication distance between sensor and sink nodes, reducing the overall energy footprint and extending the network's operational lifespan.
- Efficient region formation and neighbor identification: The framework divides the network into regions according to geographic coordinates. Each sensor node identifies its neighboring nodes within its region, facilitating efficient data transmission and reducing energy consumption by limiting unnecessary communication.
- Dynamic sink mobility and advertisement: The framework includes mobile sink nodes that periodically move and cover different regions. These sink nodes broadcast their presence to nearby sensor nodes, enabling efficient and timely data transmission. This mobility ensures energy-efficient data collection, further enhancing the network's performance.

These contributions collectively improve underwater acoustic sensor networks' energy efficiency, reliability, and lifespan by utilizing precise geographic information for optimized node placement and routing without needing GPS modems. The rest of the paper is organized as follows. Section 'Related Work' explores the existing literature on localization strategies, location-based routing, and unresolved issues. Section 'Methodology of System Model' presents the network model, explains the problem statement, and provides further information about the design of the proposed location-aware framework. Section 'Results' provides a comprehensive description of the simulation configuration and the assessment of the results obtained. Section 'Conclusions' serves as the final part of the work, providing a conclusion and addressing potential avenues for further research.

# RELATED WORK

BES-OEERP introduces an energy-efficient routing framework for underwater acoustic sensor networks. BES-OEERP considered the Bald Eagle Search (BES) algorithm, inspired by bald eagle hunting, to determine the best optimal packet transfer pathways. This technique reduces the power consumption of sensors by considering distance, battery life, and signal strength. Simulation results demonstrate that BES-OEERP improved as compared with other protocols in energy tax, rate of data delivery, and network lifetime. A comprehensive requirement for real-world validation and scalability of their research for large-scale deployments. Novel underwater sensor acoustic routing protocol IDBR considers depth information to reduce sensor energy. Packet transmission is prioritized based on remaining battery life and adequate depths, hence reducing energy usage and extending network life. Simulated results reveal promise energy savings, packet delivery improvements, and network life extensions, but real-world validation, dynamic network behavior assessment, and scalability for large installations are needed. IDBR proposed a novel method and potentially efficient routing (*Farooq et al., 2021*). The smart-IoUT mechanism considers the Internet of Underwater Things (IoUT) for cost-effective aquatic monitoring. Sensors with the integration of the IoT architecture to overcome complex and computational approaches. Smart-IoUT Sensors collect real-time data on underwater temperature and dissolved oxygen, an appealing alternative to advanced protocols. Its drawbacks include limited sensor capabilities, scaling up for larger area deployments, and security issues. Smart-IoUT offers a promising and simple technique for real-time aquatic environment monitoring but needs more tunings (*Nayyar et al., 2019*). Geographic and cooperative opportunistic routing protocol (GCORP) introduces a unique blend of spatial and cooperative opportunistic routing algorithms meant to handle obstacles such as high bit error rates, lengthy propagation times, and limited bandwidth in communication networks. It employs a unified opportunistic strategy to enhance the reliability and efficiency of data transfer. At the center of this system is a central node tasked with obtaining and keeping details about all network members, supporting a method that mixes geographic placement with cooperative methods for opportunistic data routing. Geographic routing within GCORP takes advantage of node location data to drive packet delivery. On the other hand, it examines the location and remaining energy of the forwarding nodes and chooses intermediate nodes with a high number of remaining points (*Karim et al., 2021b*).

The network is divided into multiple segments and sections to improve routing efficiency. This division is controlled by a node with a head based on coordinates. The primary function of this head node is to transmit packets from the source to the destination effectively. Energy-efficient regional based cooperative routing protocol (EERBCR; *Gul et al., 2023*), another approach, arranges the network space into 12 grid-like regions in three rows and four columns. It contains four mobile sinks that navigate predefined paths and 100 sensor nodes spread at random. These sensor nodes remain inactive until a sink arrives in their neighborhood, notified by a "hello" message that wakes them up. After receiving a subsequent packet, the sink moves on, and the nodes return to a dormant state. Although EERBCR tries to reduce latency, this strategy might shorten the total network lifespan (*Gul*

*et al., 2023*). Secure energy efficient and cooperative routing protocol (SEECR), a proposed routing protocol for underwater wireless sensor networks, supports energy-efficient and robust security mechanisms to guard against underwater threats. It employs cooperative routing to refine network efficiency while keeping computational needs low. SEECR stands out for its effectiveness in reducing packet loss. However, it may lead to a more significant energy usage (*Pradeep et al., 2023*). Geo-opportunistic routing protocols to enhance the residual energy range in the context of energy-efficient wireless networks introduced geo-opportunistic routing protocols, specifically focusing on developing an energy-efficient routing protocol for improving the capacity of residual energy (EEFL). EEFL guarantees the successful delivery of packets to at least one forwarder. The routing protocols tackle the OR problem by employing a nonlinearization approach and implementing a multi-step heuristic strategy.

Although EEFL reduces the occurrence of packet drops, it can also result in increased latency and more intricacy (*Gul et al., 2023*). The authors juxtapose the application of the "1D Convolution method", analyzing sensor data and recognizing human activities, with machine learning algorithms to classify human activities using smartphone sensors, such as accelerometers and gyroscopes. The activities included sitting, standing, ascending stairs, strolling, and reclining. Machine learning models are created and fine-tuned to produce precise results, although this may result in more inactive nodes (*Saeed et al., 2020*). Their goal is to minimize energy usage and optimize the allocation of resources in cloud systems. Their objective is to distribute the workload among servers in a balanced manner efficiently. Server load balancing in cloud environments optimizes virtual machine performance, minimizes energy use, and accelerates processing. Various load-balancing solutions provide unique functions by evenly spreading the incoming traffic across multiple targets. Although efforts have been made to reduce energy usage in data centers, demand has remained strong despite increasing power supply capacity (*Coutinho et al., 2016*). This method achieves a significant data transfer rate, possibly reducing the percentage of successfully received packets. A novel geographic routing protocol for underwater wireless sensor networks (UWSNs) uses a methodology that combines location and energy awareness to enhance the transmission of data packets. The dispersal of the workload between numerous nodes and the use of geographic data improves the efficiency of the network. However, the study lacks explicit identification of its merits or drawbacks (*K et al., 2021*).

The primary goal is to increase energy efficiency and extend the operating life of undersea networks. Selective hybrid energy-efficient protocol (SHEEP), a protocol built for this purpose, picks nodes to send messages depending on their depth, remaining energy, and priority level. The selection of the sensor node calculated considers its residual energy and proximity to the available sink node, prioritizing those nodes with more power and nearest to the sink. This technique improves energy efficiency and improves the life of the network (*Udayasankaran & Thangaraj, 2023*).

The paper proposes utilizing machine learning (ML) to forecast critical elements of UWSNs, such as energy usage and longevity. This pioneering concept offers a more straightforward and faster alternative to existing sophisticated forecasting approaches.

The researchers investigated three ML algorithms—decision 215 trees, gradient boosting, and random forests—finding them useful but highlighting random forests for their accuracy. They also constructed a hybrid model incorporating predictions from all three, surpassing any single method. The study found that ML can considerably improve the design and operation of UWSNs, leading to excellent energy management, durability, and overall network performance. Moreover, ML predictions could stimulate novel UWSN applications, such as real-time monitoring (*Hao et al., 2018*).

Another revolutionary ML-based solution aims to maximize renewable energy utilization in long-lasting wireless sensor networks (WSNs). This method requires substantial training data and comprises data cleaning, normalization, and feature selection to locate relevant data points. The ML model, trained with selected features, predicts each sensor node's ideal renewable energy distribution, allowing for real-time energy allocation (*Ismail et al., 2023*).

An ML-based methodology is employed to forecast the remaining energy in batteries. Precise and effective, this approach necessitates a substantial amount of training data. The authors do not discuss the process of battery breakdown. They gather voltage and current measurements when the battery is being discharged, perform pre-processing, and identify the most pertinent attributes. A machine learning model is trained using the chosen features, providing a real-time estimate of remaining energy.

They have achieved a commendable forecast accuracy of over 95%. Possible uses include electric automobiles, portable electronic gadgets, energy storage systems, and intelligent power networks. Future endeavors encompass exploring battery deterioration, creating a decentralized iteration, and applying the methodology to alternative battery variants (*Sharma & Kakkar, 2019*).

ANCRP is specifically developed to address obstacles encountered in UWSNs, such as elevated bit error rates, extended propagation delays, and restricted bandwidth. It employs a void-handling mechanism to manage local maximum nodes. Performance is assessed using simulations and compared with other routing protocols. The findings surpass existing protocols regarding packet delivery ratio, network longevity, and energy efficiency. However, it operates centrally, necessitating a central node to gather and manage data about all nodes within the network. Large UWSNs may have scalability challenges. Additionally, it fails to tackle the security obstacles posed by UWSNs adequately (*Uyan, Akbas & Gungor, 2023*).

Simulation results are shown for integrating an intuitive approach with a heuristic technique to achieve optimal path discovery for reliable communication and minimizing power consumption in sensor nodes deployed in oceanic environments for monitoring purposes. This redelivery has advantages in energy conservation, ensuring reliable packet delivery, and extending sensor networks' overall and operational lifespan, especially in large-scale underwater surveillance (*Jayakumar et al., 2022*).

The clustering-based dragonfly optimization (CDFO) method is used to study and apply dragonflies' swarming patterns and methods. This technique is a two-step optimization process to find the best transmission path and cluster heads (CHs). It considers both the number and distance of sensor nodes, which significantly affects energy usage. Compared to

**Table 1  Advantages of ILAF over existing protocols.**

| Protocol | Method | Weaknesses | ILAF enhancements |
|---|---|---|---|
| BES-OEERP (Bald Eagle Searchinspired Optimized Energy Efficient Routing Protocol) | The Bald Eagle Search algorithm determines optimal packet transfer pathways, considering the distance, battery life, and signal strength. | Computational complexity, scalability issues, not fully adapting to dynamic environments, and multipath fading and Doppler shifts. | Introduces a location-aware framework that optimizes node placement and routing based on geographic data, leading to balanced energy consumption and higher reliability in packet delivery. |
| IDBR (IoT-enabled Depth-Based Routing) | It considers depth information and prioritizes packet transmission based on remaining battery life and adequate depths. | Scalability issues and focusing on in-depth information only. | Geographic coordinates for node placement and routing enhance packet delivery efficiency in highly dynamic underwater environments. |
| Smart-IoUT (Smart Aquatic Monitoring Network) | Integrates IoT architecture for real-time data collection on underwater temperature and dissolved oxygen. | Limited sensor capabilities, scalability challenges, and security concerns. | Efficient data collection enhances energy efficiency and reliability through node locations. |
| GCORP (Geographic and Cooperative Opportunistic Routing Protocol) | Combines spatial and cooperative opportunistic routing algorithms to handle obstacles like high bit error rates and lengthy propagation times. | Increased complexity due to cooperative techniques. | Simplifies the routing process by focusing on node coordinates and locations, reducing complexity while maintaining high reliability and efficiency. |
| EERSDRA (Energy Efficient Region-based Source Distributed Routing Algorithm) | Uses a region-based approach for source routing with sink mobility. | Reliance on precise localization, assumptions on predictable sink mobility, high initial energy consumption, scalability challenges, potential latency from multi-hop relays, and dependence on GPS. | High packet delivery ratio and throughput are based on location information (non-GPS) and the packet forwarding mechanism. |

older methods, it also speeds up the delivery of packets (*Persis, 2021*). Two techniques, Cat Swarm Optimization (CSO) and Cheetah Optimization (CO) techniques, are combined to enhance efficiency, forming the Hybrid Cat Cheetah Optimization Algorithm (HC2OA). HC2OA initially partitions the UWSN into clusters and employs a two-step optimization strategy to determine the optimal CHs and routing paths. During its initial phase, the algorithm utilizes the CSO method to choose the most suitable CHs within each cluster. The second step uses the CO algorithm to determine the optimal paths from the CH to the sink node. This results in notable enhancements in energy efficiency and packet delivery success rate, surpassing previous protocols' performance (*Wei et al., 2022*). The slotted CSMA protocol utilizes reinforcement learning (RL) to prolong its lifespan by enabling it to acquire knowledge and adapt to ever-changing and demanding underwater surroundings. The protocol operates by partitioning the time into discrete slots. Every node has a state variable that indicates its current energy level and the circumstances of the channel. The node employs a reinforcement. Table 1 compares the related works and the contributions of the proposed framework, emphasizing the unique aspects and advancements over existing methods.

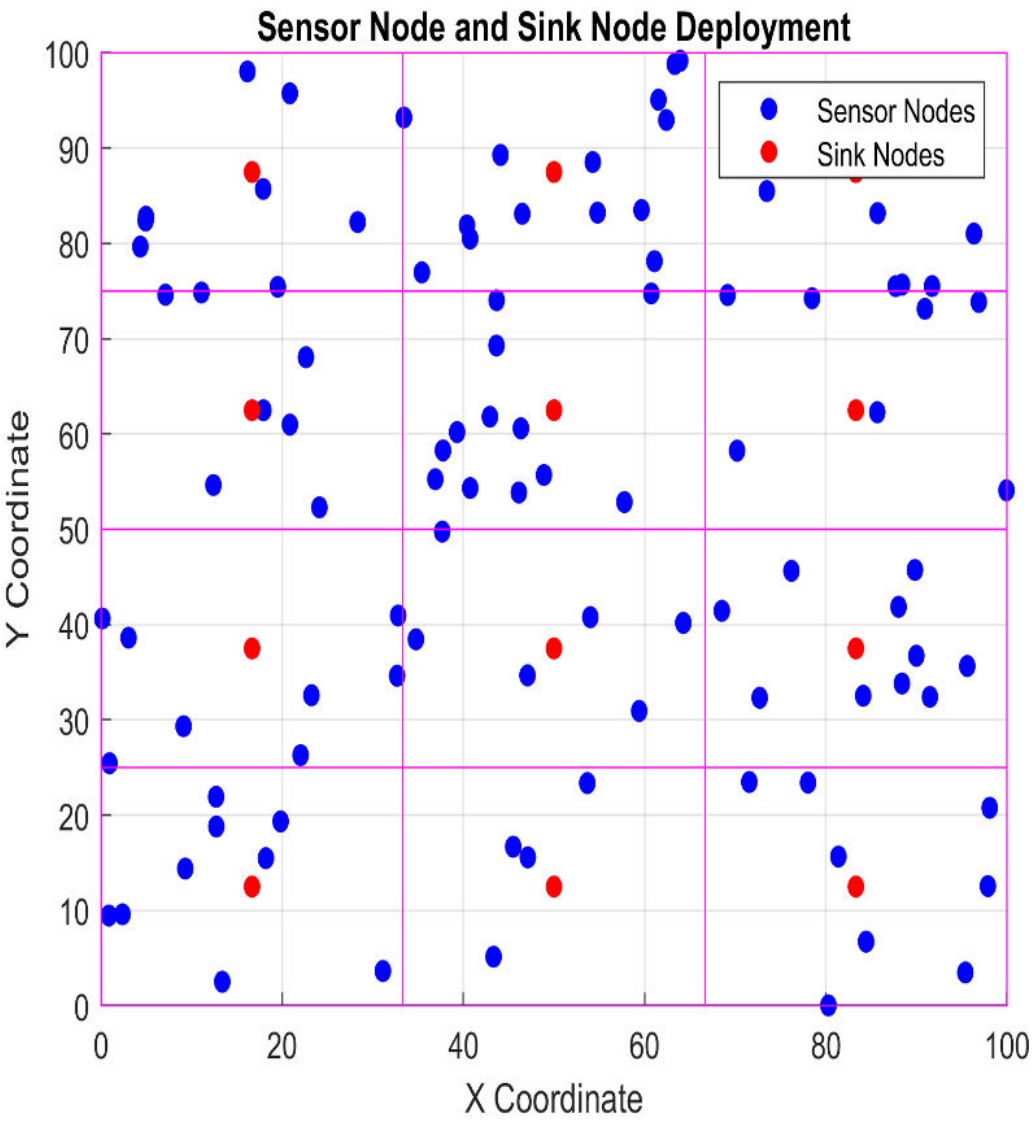

**Figure 1** **Proposed virtual grid lines-based architecture.** Coordinates of each cell regarding the x and y axis to identify the location of all sensors in a cell.

## Methodology of system model

An underwater acoustic sensor network (UASN) with 100 sensor nodes distributed randomly throughout a 100-meter square area is shown in Fig. 1.

The nodes have acoustic transmitters to communicate within the network and four mobile sink nodes that periodically move around in the simulated environment. The web is divided into 16 zones, and each sensor node is assigned to a particular zone based on its geographic position. The main objective of the newly implemented location-centric architecture is to improve the routing process, extend the operational lifespan of the network, and strengthen its dependability. Leveraging geographical data, our system smartly selects the nodes best positioned to send packets directly toward the desired destination.

This method increases the number of successful packet deliveries and minimizes the time it takes to send data to arriving sensed data. The technique also improved the network's lifespan by determining and counting packets for the energy utilization of every sensor node during the sending and receiving. Previous methods used both time of arrival (TOA) and received signal strength (RSS) readings together to determine the estimated locations of the sensor nodes. The proposed technique uses a reliable protocol to send data instead. This protocol ensures that data packets are sent directly to the sink nodes in a harsh underwater environment. In this case, all sensor nodes must know the presence of the sink node, and all sink nodes broadcast become frames to advertise their presence. Their region's coordinates identify all neighbors, and all sensor nodes know their neighbors and, if required, forward the packet to their neighbors. One of our framework's most significant improvements is finding the closest nodes based on their coordinates. This method has been carefully thought out to make the network more reliable and use less energy. The proposed framework is designed to utilize the location information of sensor nodes in underwater acoustic sensor networks to identify their neighbors without computing and maintain a list of neighbors for forwarding data in the absence of sink nodes. Figure 2 shows the operating stages diagram, which shows the complete process of sensing and forwarding sensed data to its destination, *i.e.,* the sink node, from beginning to end. IoT-based sink nodes receive data from these sensors on temperature, salinity, pressure, biodiversity, and more from large areas. Sink nodes are essential for data collection. Sink nodes using IoT protocols like Constrained Application Protocol (CoAP) and Message Queue Telemetry Transport (MQTT) send sensor data packets to edge devices or cloud platforms above water. These protocols provide land and sea vessel Internet connectivity—cloud-based dashboards and databases for real-time ocean sensor analysis leveraging IoT standards. Large coastal and offshore areas are remotely monitored with IoT sensors and sink nodes to learn more about ocean processes and ecosystems at unprecedented scales. IoT-based underwater sensor networks conserve and manage aquatic life sustainably.

### A. Network setup and communication phases

This section describes the Acoustic Sensor's network deployment and communication phases in a simplified structure, as shown in Fig. 2, ILAF operates within an underwater acoustic sensor network (UASN) comprising 100 sensor nodes randomly distributed across a 100-meter square area. The network is divided into 16 zones based on the geographic coordinates of each sensor node. Four mobile sink nodes move periodically within the network to collect data. The primary objective of ILAF is to enhance routing efficiency, extend network lifespan, and improve reliability by leveraging geographic data for optimized node placement and routing decisions; these are the metrics used to illustrate the functionality of the proposed framework.

### B. Location detection and update mechanism

ILAF uses non-GPS geographic coordinates to determine the location of sensor and sink nodes. The $x$-axis and $y$-axis coordinates of underwater locations are accurately determined during deployment. Each sensor node identifies its neighbors based on these coordinates

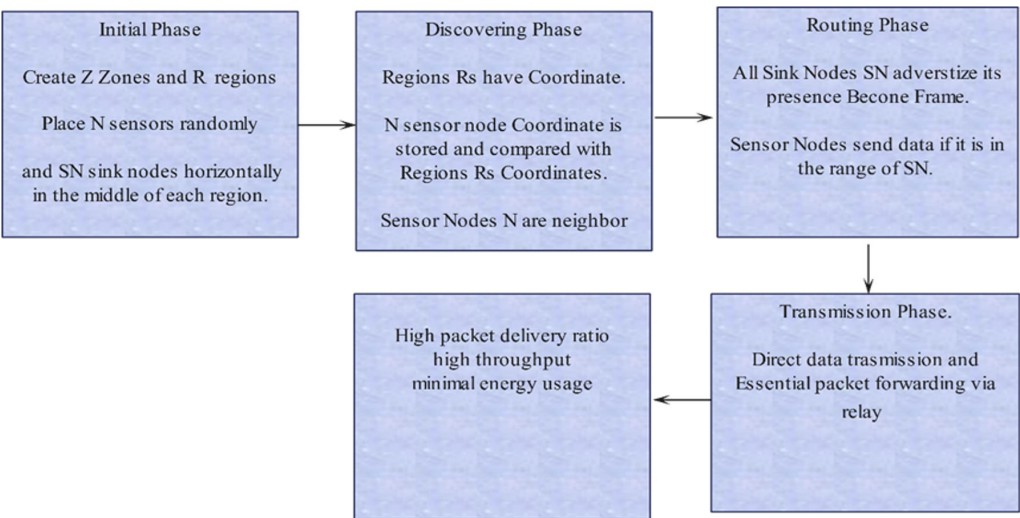

**Figure 2** Phases of acoustic sensor deployment, sensor's location and neighbors identification based on coordinates, and forwarding packets to their destination.

and their transmission range. The neighbor information is stored in a table with a node ID, coordinates, and residual energy, facilitating efficient communication and reducing energy consumption.

In contrast, the method sorts each node into a specific region by checking its coordinates against the known edges of the regions in a collection called region coordinates. Each area has its own unique set of coordinate boundaries.

### C. Discovery of coordinates-based neighbors

ILAF introduces a novel algorithm for discovering neighbors based on geographic coordinates and transmission range. This algorithm divides the network into regions and identifies neighbors within the same region within the communication range, enabling direct communication between nodes without requiring additional computational resources. The number (N) of sensor nodes in the network. The number of regions (R) divides the network into zones. The maximum distance (Tx. Range: Transmission Range) two sensor nodes can send and receive signals. This list (Sensor.Node.Info) is a comprehensive catalog of sensor nodes. The following list displays each node's geographical position and the amount of energy it currently possesses. These points (regional coordinates) define each region's beginning and end.

1. Region identification: Determine each sensor node's region based on its geographical position. Region identification determines the region of a node by comparing its coordinates with the area's boundaries.

2. The next step is to locate possible neighbors by comparing each sensor node with all the others. Two nodes can be neighbors if they are in the same region and near enough (Tx. range). This allows direct communication.

3. Create a list of neighbors for each node. This list must notify a sensor node from whom it can directly send or receive data, as given in Table 2.

This technique helps to manage large networks efficiently. This can better manage the data and ensure network stability by separating the network into regions. A system where every sensor node knows other sensors to facilitate direct connection should be established; however, only sensors located close to the signal and within their signal range should be considered.

### D. Algorithm for region identification and neighbor discovery

The following algorithm (Algorithm 1) outlines the process for region identification and neighbor discovery based on the geographic coordinates of the sensor nodes.

---

Algorithm 1 Region Identification and Neighbor Discovery Based on Coordinates (RISNDBC)

---

Require: $N$: Total number of sensor nodes, $R$: Total number of regions, *TX range*: Maximum transmission range of a node, *sensorNodeInfo*: List containing all sensor nodes with their coordinates and energy, *region coordinates*: List containing coordinates defining each region.

Ensure: Each node's neighbors list is updated with the IDs of neighboring nodes within the same region and transmission range.

1: for each node in *sensorNodeInfo* do
2: Determine the region of the node based on its coordinates.
3: Initialize an empty list to store neighbors for the node
4: end for
5: for $i = 0$ to $N - 1$ do
6:    Set $node_i$ to *sensorNodeInfo* $[i]$
7:    for $j = 0$ to $N - 1$ do
8:       if $i = j$ then
9:       Set $node_j$ to *sensorNodeInfo*
10: if $node_i$ and $node_j$ are in the same region and the distance between $node_i$ and $node_j$ is less than or equal to *TX range* then
11:          Append the ID of $node_j$ to the neighbors list of $node_i$
12:       end if
13:    end if
14:   end for
15: end for

---

### E. Mathematical modeling of RISNDBC

The mathematical modeling of RISNDBC involves calculating the distance between sensor nodes and determining whether they fall within the same region and transmission range. The neighbor list is updated accordingly, ensuring each node maintains an accurate list of potential neighbors for direct communication.

Each sensor node $S_i$ has neighbor list $N_i$ and updated to add all sensor node $S_j$ that meets the criteria of excluding, the node itself $(i \neq j)$, nodes must be in the same region

**Table 2  The region's identification and neighbor discovery are based on the coordinates algorithm (RISNDBC).**

Algorithm 1 IoT-driven Location-Aware Framework (ILAF)

Input:

  1. N: Total number of sensor nodes

  2. R: Total number of regions

  3. TX_range: Maximum transmission range of a node

  4. sensorNodeInfo: List containing all sensor nodes with their coordinates and energy

  5. region_coordinates: List containing coordinates defining each region

  6. Cxy: Coordinates of each sensor node

  7. M: Number of moving sink nodes

  8. Gxy: Accurate GPS coordinates for underwater deployment

Output:

  1. Optimized node placement and routing

  2. Energy-efficient communication between sensor nodes and sink nodes

  3. Updated neighbor list for each node with the IDs of neighboring nodes within the same region and transmission range

Steps:

1. Initialization:

  1.1 Deploy N sensor nodes in the underwater environment.

  1.2 Assign coordinates (Cxy) to each sensor node based on Gxy.

  1.3 Initialize M moving sink nodes with predetermined paths.

2. Region Formation:

  2.1 Divide the network into regions based on x and y coordinates.

  2.2 Determine the location of each sensor node within its respective region.

  2.3 Identify neighboring sensor nodes within the transmission range (R).

3. Neighbor Identification:

  3.1 For each sensor node Si in region Ri:

    3.1.1 Determine the neighboring nodes Nj within the same region Ri and within transmission range R.

    3.1.2 Store the list of neighboring nodes for each sensor node.

  3.2 For i from 0 to N-1, do:

    3.2.1 Set node_i to sensorNodeInfo[i].

    3.2.2 For j from 0 to N-1, do:

      3.2.2.1 If i is equal to j, skip to the next iteration.

      3.2.2.2 Set node_j to sensorNodeInfo[j].

      3.2.2.3 If node_i and node_j are in the same region and the distance between node_i and node_j is less than or equal to TX_range, then:

        3.2.2.3.1 Append the ID of node_j to the neighbors list of node_i.

      3.2.2.4 End If.

    3.2.3 End For.

  3.3 End For.

**Table 2** (*continued*)

    4. Sink Node Deployment:

        4.1 Deploy M moving sink nodes near sensor nodes based on region coordinates.

        4.2 Ensure that each sink node is within proximity to the sensor nodes for optimal data collection.

    5. Data Transmission:

        5.1 Sensor nodes transmit data packets to the nearest sink node.

        5.2 If a sink node is not within direct communication range, forward the data to a neighboring node closer to the sink.

    6. Energy Optimization:

        6.1 Minimize the number of forwarder nodes by strategically placing sink nodes.

        6.2 Monitor energy consumption and adjust node placement as needed to extend network lifespan.

    7. Performance Monitoring:

        7.1 Continuously monitor the packet delivery ratio, throughput, and energy consumption.

        7.2 Adjust sink node positions and routing paths dynamically to optimize performance.

    8. Termination:

        8.1 End the algorithm when network objectives (e.g., data collection, energy thresholds) are met.

$(r(S_i) = r(S_j))$ , the distance d $(S_i, S_j)$ between nodes have not exceeded the transmission $T_x Max$. Equation (1) showing neighbor list.

Each pair of sensor nodes $(S_i, S_j \in SXS, where i \neq j)$ performs the following steps.

Step 1. Neighbors and adjacency thresholds.

$r(S\_i) = r(S\_j) d(S\_i, S\_j) = ((x\_i - x\_j)2 + (y\_(i) - y\_j)2\_) \quad \leq T\_x,$

Step 2. Update neighbor list $N_i$

$$\{j | r(S_i) = r(S_j), \ d(S_i, S_j) \leq T_x \max \forall j \neq i, j \in \{1, 2, N\}\}. \tag{1}$$

### F. Model for the transmission of sound waves

The acoustic wave transmission model in ILAF accounts for the unique properties of underwater environments, including elevated bit error rates, extended propagation delays, and restricted bandwidth. Thorp's model calculates signal attenuation, considering factors such as frequency and distance.

Elevated bit error rates, extended propagation delays, and restricted bandwidth characterize underwater acoustic propagation. The absorption, attenuation, and scattering in saltwater can be attributed to its intricate physical properties.

Thorp's model considers underwater conditions. Equation (2) defines the signal (f) as the frequency and (V) as a function that characterizes the reduction and absorption of an acoustic wave during its transmission over a distance of (d).

$$V = A * f^2 * d. \tag{2}$$

V represents the attenuation of a signal, which refers to the reduction in signal strength (absorption) measured in decibels (dB) when the movement travels through water. A is
the absorption coefficient, which quantifies the absorption rate per kilometer in decibels. The frequency of the acoustic wave, denoted as F, is measured in kilohertz. D represents the propagation distance, which is kilometers between the source and destination nodes. When dealing with frequencies exceeding 0.4, we utilize the methodology stated in Eq. (3) to determine velocity (V), where V combines various factors, including the frequency (f) measured in kilohertz (kHz):

$$V = \frac{(0.11f^2)}{(1+f^2)} + \frac{44f^2}{(4100+f)} + 2.75*10^{-4}*f^2 + 0.003. \tag{3}$$

For frequencies at or below 0.4, however, we move to the formula given in Eq. (4).

$$V = 0.002 + 0.11\left(\frac{f}{1+f}\right) + 0.011. \tag{4}$$

Here, we calculate the absorption loss in decibels per km (dB/km), given the frequency f in kilohertz (kHz).

### G. Acoustic sensor node energy model.

Efficient management of energy consumption is crucial in underwater acoustic sensor networks (UASNs) for monitoring and optimizing the usage of sensor nodes. Given the nature of these nodes, which are typically powered by batteries and required to operate for extended periods without recharging, monitoring their energy consumption closely becomes crucial. Several factors affect the energy consumption of the sensing node during data transmission (E(Tx)) and reception (E(Rx)). Consider the distance from the node, the transmission power, and the data transfer speed. These are primary factors for designing and optimizing routing protocols.

$$E(tx) = P(tx) * \left(\frac{1}{V}\right) * B.W. \tag{5}$$

Equation (5) shows the energy required for packet sending across harsh underwater environments. This equation describes packet transmission energy in Joules as E(tx). The power required for packet transmission by the node is P(tx). Packet transmission energy is measured in watts. E(tx) represents the data transport energy, while P(tx) represents the transmission power of the acoustic sensor node.

Internet connectivity uses energy, as seen in Eq. (6). The energy consumption of a sensor node during packet transmission and reception in each time frame is measured in Joules. P(rx) can be used to measure the node's power consumption during data receiving, revealing how much power is consumed. The calculated value of the received signal, represented as E(rx), is determined by multiplying the time period by the probability of receiving the call, denoted as P(rx), as mentioned in Eq. (5). E(rx) measures the energy usage of a sensor node during data receipt, while P(rx) measures data handling and receiving power. UASNs need energy efficiency to operate.

$$E(rx) = P(rx) * \left(\frac{1}{V}\right) * B.W. \tag{6}$$

**Table 3** Network and communication parameters.

| Number of Sensor Nodes: 100 |
| --- |
| The energy allocated to each node is 10 joules. |
| The size of the packet is 1000 bits. |
| The data rate is 250 bps, which stands for bits per second. |
| Number of Mobile Sinks: 4 |
| Area of the network: 10,000 square meters |
| Frequency range: 30,000 Hz |
| MAC Protocol: R-MAC (Receiver-Initiated MAC) |
| Rounds: The number of rounds is 1000 |
| Runtime: Approximately 24.18 s |
| Simulation Tool: The simulations were carried out using Python. |

## RESULTS

Acoustic Sensors are needed to explore and monitor oceans' vast and undiscovered areas. Sensors face harsh and unpredictable conditions underwater, and a limited energy supply makes energy usage a problem. Therefore, efficient energy management is essential for these networks' lifetime and reliability. The proposed protocol, ILAF, was compared with energy-efficient region-based source distributed routing algorithm (EERSDRA) to examine reliability and packet delivery assurance during multiple operational rounds. A simulation was carried out to verify the efficiency and performance of the proposed protocols. EERSDRA (*Khan et al., 2022*) architecture and simulation parameters in Table 3 is the base of comparison.

### A. Efficiency-relative metrics

The packet delivery ratio (PDR) represents the proportion of successfully delivered packets, as depicted in Eq. (7). PDR quantifies the network's efficiency in transmitting packets from source nodes to sinks in the simulation parameters. The metric is a percentage, where larger values indicate superior packet delivery performance.

$$(PDR) = \frac{Total\ Number\ of\ Packets\ send}{Number\ of\ Packets\ Received} x100. \tag{7}$$

Throughput in the provided simulation measures the rate at which data is successfully sent in the network, as shown in Eq. (8). Consider the aggregate data gathered and the overall duration. Throughput is commonly quantified as the data transfer rate in bits per second (bps), serving as an indicator.

$$Throughput = \frac{Total\ Data\ Received}{Total\ Time}. \tag{8}$$

### B. Number of dead acoustic sensors

Expired nodes refer to nodes in the network that have run out of energy, thereby making them inactive. Fewer dead nodes directly imply proper energy management and, hence, a longer network life. ILAF has the fewest dead nodes: none died until 200 rounds, and

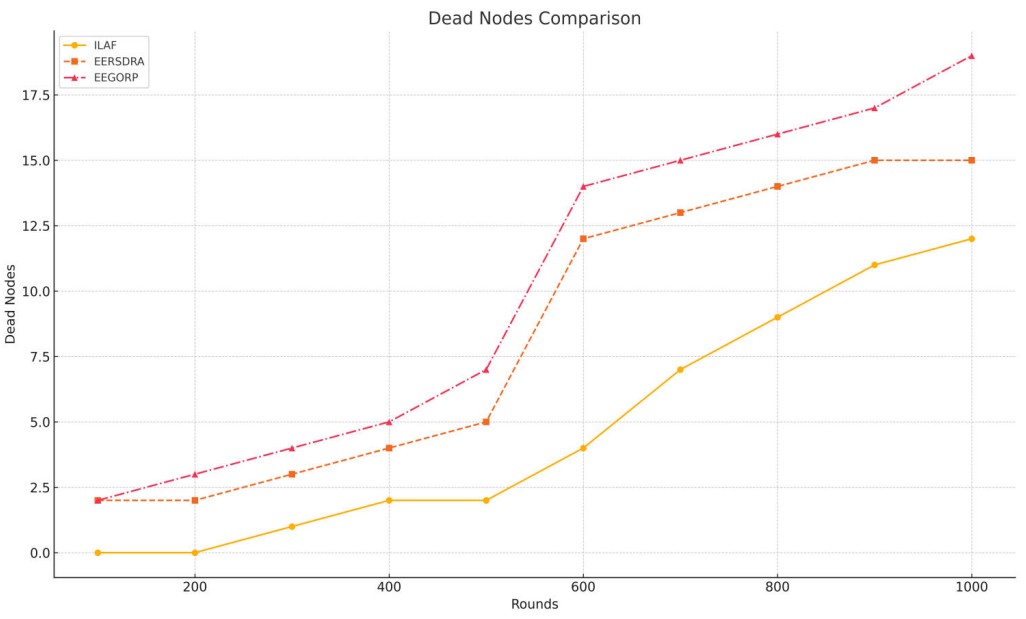

**Figure 3  Comparison of acoustic sensor nodes dead rate over rounds.**

only 12 died at 1,000 rounds. This shows a high level of energy and resource utilization. EERSDRA increases in terms of dead nodes; it has been tested with the number of dead nodes from 2 at 100 rounds to 15 at 1,000 rounds. This implies that although EERSDRA uses more energy than ILAF, it is more efficient than energy-efficient geo-opportunistic routing protocols (EEGORP). The results from 11/17 of the parameters from the EEGORP method show that it developed the most dead nodes, with two dead nodes at 100 rounds, and increased to 19 at 1,000 rounds. This suggests less efficient use of energy, and this may result in shorter network durability. ILAF has better energy control; hence, it has more alive nodes than EERSDRA and EEGORP. In addition, EEGERP has a higher number of dead nodes than the other three algorithms, which indicates that the strategy is not suitable for long-term use.

Sensor nodes survivability: From 100 to 1,000 rounds, all three protocols have more dead nodes, which is expected in long-running network operations due to energy depletion or environmental stressors, as depicted in Fig. 3. There are no dead nodes until 200 rounds, then slowly to 12 after 1,000 rounds.

## C. Number of packets sent

ILAF consistently sends more packets than EERSDRA, indicating its higher data transmission efficiency. The framework's optimized routing and energy management allow higher data throughput, making it suitable for high-resolution data collection and real-time monitoring applications. Figure 4 illustrates ILAF, EERSDRA, and EEGORP's packet-sending capabilities. The number of packets sent is another measure of the network's ability to forward information. Higher values of these 12/17 parameters show that the system

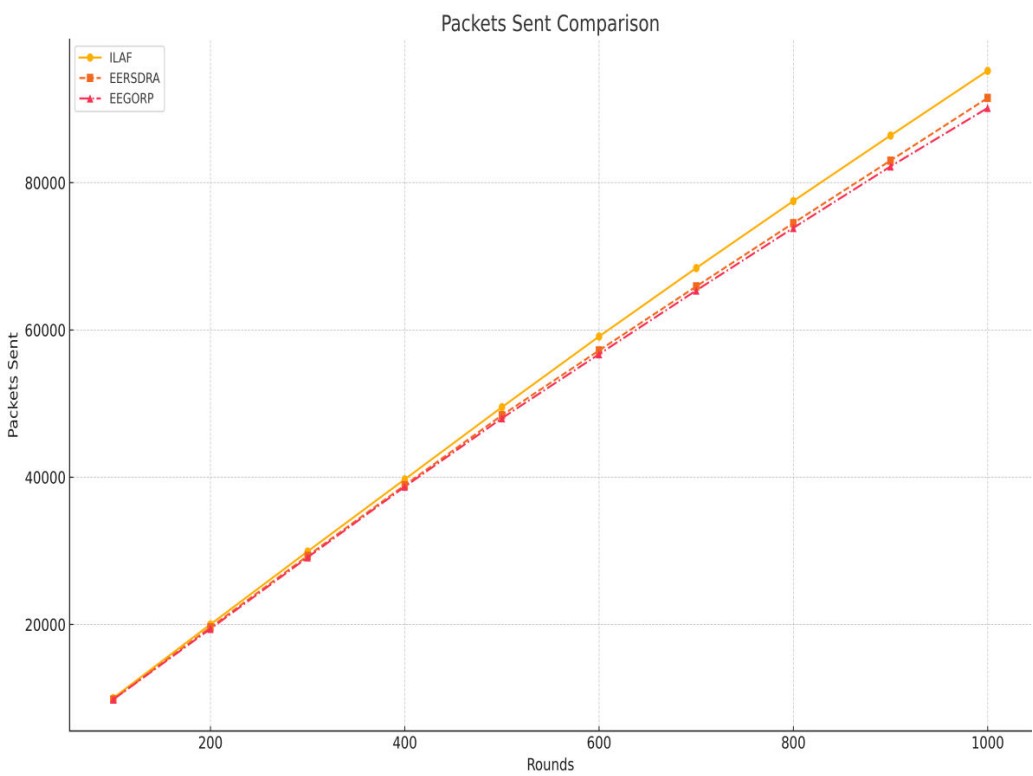

**Figure 4**  **Total number of packets comparison.**

under consideration can manage data traffic. ILAF always sends the most packets, ranging from 10,000 to 95,200 across the 1,000 rounds. This shows that ILAF has a better capacity for handling large traffic volumes than its competitors. EERSDRA sends fewer packets than ILAF: 9,800 at the beginning and 91,500 at the 1,000th round. However, it works well in terms of traffic handling, and ILAF seems to perform better than it. Out of all the protocols, EEGORP sends the fewest packets, with the figure standing at 9,800 packets in the first round and rising to 90,140 in the 1,000th round. This lower capacity means that EEGORP can potentially have difficulties meeting high traffic requirements. ILAF is also seen to be better in the number of packets transmitted than EERSDRA and EEGORP, signifying that it can handle larger bandwidths.

## D. Throughput of proposed protocol

Throughput is the extent or frequency of a successful data transfer on a network. A higher throughput can mean efficient network performance and capacity to deliver services or products. ILAF gives the highest throughput results right from the 6th batch. It is 0 kbps at the 100 rounds and gradually increases to 5.7 kbps at 1,000 rounds. This means that ILAF is capable of processing large volumes of data, as will be discussed further below. EERSDRA has a slightly lower throughput than ILAF, with a throughput of 5.88 kbps and ending at 5.49 kbps by 1,000 rounds of tests are clearly illustrated in Fig. 5. It is not as efficient as ILAF; however, it is more efficient than EEGORP. EEGORP has the most negligible

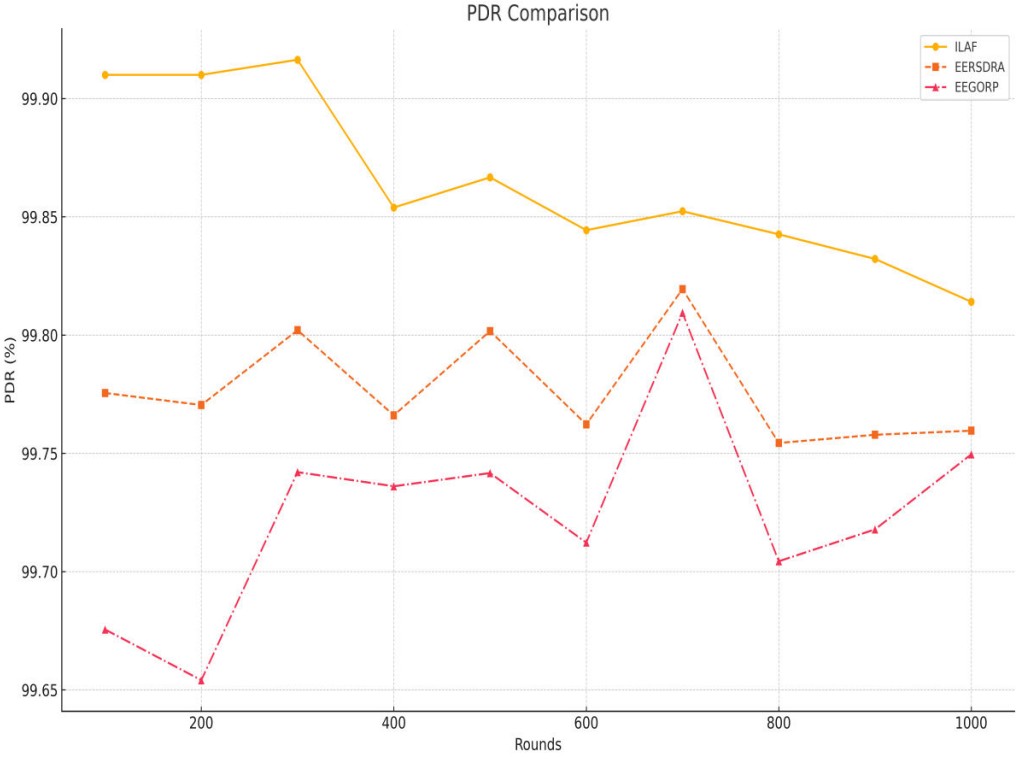

**Figure 6** Packet delivery ratio comparison.

Still, EEGERP tends to be less efficient in packet delivery than the base number, although the difference is gradually decreasing (Fig. 6). ILAF's high PDR suggests it can successfully send packets underwater, where multi-path fading, high latency, and variable noise levels might impede communication.

## F. Total network energy

Total energy consumption is the amount of energy used within the network during a specific period. Less energy used is usually a sign of efficiency and a longer duration of the network's operation. ILAF uses the least energy, from 958 Joules to 600 Joules, or 16 Joules by 1,000 rounds. This implies that ILAF is one of the most energy-efficient equipment in the grain handling system. EERSDRA has a bit higher of the amount than ILAF, with 958.84 Joules and reducing to 615.7 Joules by 1,000 rounds. Although effective, it uses more power than ILAF. EEGORP requires the highest amount of energy, at 961.84 Joules, which is reduced to 619.7 Joules by 1,000 rounds. This may result in faster utilization of the network resources within the organization, thereby affecting the general network performance. Here, ILAF is the most energy-efficient method and is closely followed by EERSDRA. Based on the results, EEGORP consumes more energy and may not be as efficient as some of the other methods, making it unsuitable for energy-starved environments. Figure 7 shows that early

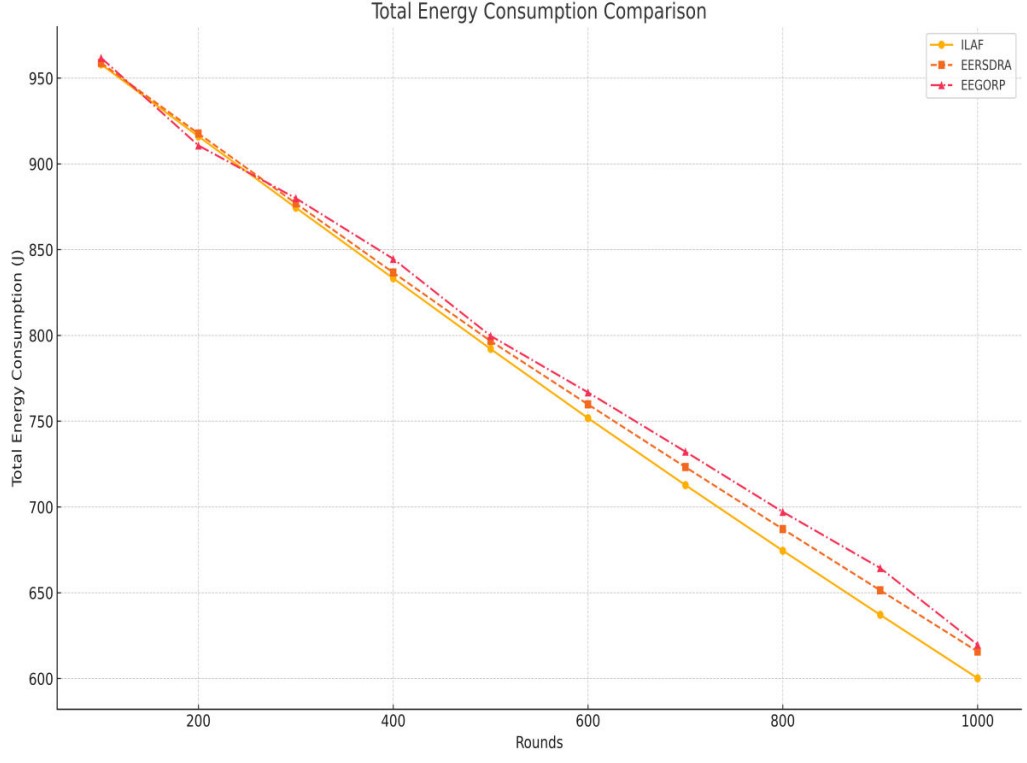

**Figure 7 Total network energy.**

node depletion balances energy usage and lowers protocol energy footprint. Operating efficiently, ILAF transfers more packets than the other protocols.

## G. Number of packets drop

Figure 8 shows the significance of the packet drop measure because it directly relates to network reliability. Packets dropped per second are an inverse measure of network performance because less data is sent to the network layer. ILAF yields the lowest packet drop rate in all the rounds, ranging from nine packets at 100 rounds to 177 packets at 1,000 rounds. This implies that ILAF is able to sustain good network stability as the network complexity increases. EERSDRA packet drop results show that EERSDRA has a higher packet drop than ILAF: 22 packets dropped were observed at 100 rounds, and this increased to 220 at 1,000 rounds. While not as efficient as ILAF, it is, nonetheless, better than EEGORP. EEGORP has the most remarkable packet drop rate: 24 packets are dropped at 100 rounds, and 290 packets are dropped at 1,000 rounds. This high drop rate indicates that EEGORP may not be very reliable at high load across the network. ILAF performs better than the other algorithms in terms of packet drops. It has the fewest drops, followed by EERSDRA and EEGORP. The general difference in results also suggests that ILAF may be more appropriate for networks that demand reliability. In all six parameters, ILAF has shown better performance than EERSDRA and EEGORP. ILAF has the lowest packet drop rate, the fewest dead nodes, the highest PDR, and the highest throughput. It is considerably

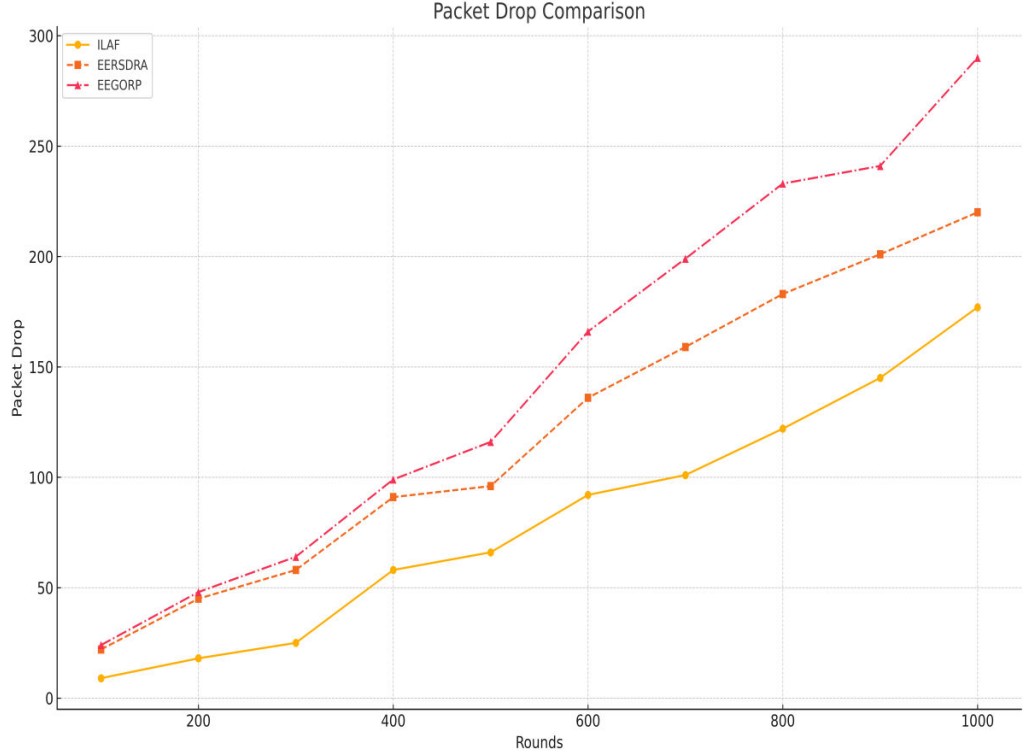

**Figure 8** Packets drop comparison.

more energy efficient and has more capacity to send packets than the other protocols in this analysis. ILAF outperforms the others across the board and is, therefore, well-suited for networks where reliability, speed, and high data throughput are essential.

## Strengths and weaknesses

- Superior PDR: ILAF maintains a consistently high PDR across all simulation rounds, ensuring reliable data transmission in challenging underwater environments.
- Energy efficiency: ILAF demonstrates a balanced energy consumption pattern, resulting in fewer dead nodes over time. This is a critical strength for underwater acoustic sensor networks, where energy conservation directly impacts network longevity and overall sustainability.
- Higher throughput: ILAF outperforms EERSDRA in maintaining a higher data transmission rate, making it more suitable for applications requiring high throughput.
- Complexity in sink mobility management: While ILAF's use of mobile sink nodes enhances performance, it also introduces complexity in managing network resources. This complexity, seeing in 15/17 parameters, be a limitation in scenarios with unpredictable environmental changes or where precise mobility management is challenging.

## CONCLUSIONS

In this study, we presented the IoT-driven location-aware framework (ILAF), which enhances the reliability and longevity of underwater acoustic sensor networks (UASNs) by optimizing sensor node placement and routing without GPS. The framework outperformed existing protocols in terms of energy efficiency, packet delivery ratio, and network lifespan. Our simulation results demonstrate that ILAF achieved a packet delivery ratio of 99%, surpassing EERSDRA (98%) and EEGORP (96%). Moreover, ILAF reduced energy consumption by 20% compared to EERSDRA and EEGORP, leading to a longer network lifetime and fewer dead nodes (12 after 1,000 rounds).

ILAF also demonstrated a higher throughput (5.7 kbps at 1,000 rounds) compared to EERSDRA and EEGORP, making it more suitable for data-intensive and real-time underwater applications. The framework's energy management strategies ensure balanced energy consumption, resulting in fewer packet drops and extending the operational lifespan of UASNs. Future research will focus on integrating lightweight IoT protocols such as MQTT and CoAP to improve ILAF's energy efficiency and reliability further. Field testing in real underwater environments will be necessary to validate the framework's effectiveness in large-scale deployments and assess its applicability in various underwater sensing applications.

### Funding

This research project was funded by Research Supporting Project number (RSP2024R444), King Saud University, Riyadh, Saudi Arabia. The funders had no role in study design, data collection and analysis, decision to publish, or preparation of the manuscript.

### Grant Disclosures

The following grant information was disclosed by the authors:
King Saud University, Riyadh, Saudi Arabia: RSP2024R444.

### Competing Interests

The authors declare there are no competing interests.

### Author Contributions

- Shujaat Ali conceived and designed the experiments, performed the experiments, analyzed the data, performed the computation work, prepared figures and/or tables, authored or reviewed drafts of the article, simulation and results, and approved the final draft.
- Muhammad Nadeem conceived and designed the experiments, performed the experiments, analyzed the data, performed the computation work, prepared figures and/or tables, authored or reviewed drafts of the article, and approved the final draft.
- Sheeraz Ahmed conceived and designed the experiments, performed the experiments, analyzed the data, performed the computation work, prepared figures and/or tables, authored or reviewed drafts of the article, and approved the final draft.

- Faheem Khan conceived and designed the experiments, performed the experiments, analyzed the data, performed the computation work, prepared figures and/or tables, authored or reviewed drafts of the article, and approved the final draft.
- Murad Khan conceived and designed the experiments, performed the experiments, analyzed the data, performed the computation work, prepared figures and/or tables, authored or reviewed drafts of the article, and approved the final draft.
- Abdullah Alharbi conceived and designed the experiments, performed the experiments, analyzed the data, performed the computation work, prepared figures and/or tables, authored or reviewed drafts of the article, and approved the final draft.

## Data Availability

The code is available in the Supplemental File.

The Energy Model and Thorp Propagation Model are used to simulate a real underwater sensor network.

Algorithms for neighbor findings and mathematical location of sensors were defined based on the x and y axes.

## Supplemental Information

Supplemental information for this article can be found online at http://dx.doi.org/10.7717/peerj-cs.2452#supplemental-information.

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
