# Peer review of "Acoustic Sensors data transmission integrity and endurance with IoT-enabled location-aware framework"

_PeerJ Computer Science, doi:10.7717/peerj-cs.2452_

## Round 0.1 · original submission · Major Revisions

The reviewers have raised several concerns that need to be addressed to improve the clarity and rigor of your study.

In what follows are some comments from the Editor about the work:

- Although a novel combination might be allowed, it is necessary to highlight the contribution of such a combination from both methodological and empirical perspectives. Also, it is required to provide technical details of the proposed methods as much as possible and in-depth explanations of method selections.

- The innovation of the paper seems limited. The proposed method is a straightforward combination of existing techniques, which makes it less innovative. Also, more details about the proposed method should be provided.

- I suggest adding a clear research objective or research questions in the introduction section and specifying what the main research problem or hypothesis is addressed

- I recommend proofreading to make reading smoother.

- There are several papers that have addressed similar problems, but it is necessary to further highlight the novelty between the proposed study and the related literature.

- Starting from the previous works, I suggest introducing a table to summarize the most recent works and to highlight the novelty of the proposed work.

- Please add more recent references. Certainly, there has been more recent (within the last two years) research on this topic published in information science and/or computer science outlets. An academic search on the topic (using keywords from the manuscript’s title) shows that there is recent work in this area. Therefore, authors must update their literature review.

- There needs to be an explicit research objective(s) and/or research question(s) stated, preferably as a separate section. This helps readers find out what the research is trying to address.

- The reference list needs tidying up, as there are references missing items or formatting issues. Please be consistent with the formatting and use some standard formatting style.

- The evaluation is weak. Please consider using a more convincing way to evaluate the proposed method, e.g., using more datasets.

- The discussion of the results does not highlight the strengths and weaknesses of the proposed approach.
* * *
We encourage you to carefully consider the detailed comments and suggestions provided by the reviewers, which are attached to this letter. Addressing these points thoroughly will significantly strengthen your manuscript.

Reviewer 1 ·

Basic reporting

All the figures in the manuscript are blurry and of low quality. The authors should redraw these figures for clarity. Moreover, the authors should use more precise language in the entire manuscript.

Experimental design

Figure 3 to Figure 8 should include 100 rounds mark on x axis.

Validity of the findings

Details included in the results section are limited. The authors should include some more technical details in the results section of the manuscript.

Additional comments

no comment.

Reviewer 2 ·

Basic reporting

Paper idea seems fine and updated but it requires overall revision to address the following concerns.

The authors must divide the large paragraphs into multiple small paragraphs (where applicable)

Some abbreviations are used before defining their full forms, e.g. GPS, IDBR, etc. Whereas some abbreviations are not defined, ref. section: Related Work. Also some are defined multiple times.

The paper has basic typing and drafting issues. The whole paper requires a thorough revision to rectify such issues and errors.

Experimental design

Can 100 nodes and 4 sink nodes are sufficient to cover whole area of 10 sq kilometer?

Authors have mentioned only few simulation parameters (Table 2), some important parameters are missing, for example, simulation tool, runtime, rounds, MAC protocol,

Validity of the findings

Very few performance metrics are utilized to evaluate the proposed scheme

It would be more better, if the proposed scheme is evaluated with more recent and relevant UWSNs routing protocols

Additional comments

Authors must clearly describe the novelty and contribution in bullet format.

Extra details are required for the terms: CoAP and MQTT.

Authors are suggested to provide more details for the network division.

Reviewer 3 ·

Basic reporting

Formal results should include clear definitions of all terms and theorems, and detailed proofs.

Experimental design

The proposed methodology is not well-explained. should expand to provide a detailed description of the approach.

Validity of the findings

The discussion on results has to be deepened in order to better highlight the advantages provided by the proposed approach.

Additional comments

The Proposed Framework maximized efficiency by identifying neighbors of sensor nodes within a region without doing computations. This paper deals with a hot area of investigation at the moment. At this p point, I recommend these parts be revised again and expressed more analytically in a revised version, as this current work is worth publication.

-The introduction effectively conveys the motivation for this study; however, the specific contributions are not clearly articulated.

-The related works section is well-constructed, but Table 1 lacks a comparative summary of the contributions of this study against existing literature and highlights how this work advances the field.

-The proposed methodology is not well-explained. should expand to provide a detailed description of the approach.

-The resolution of figures must be improved to ensure clarity.

-Some equations in the mathematical modeling section are not legible due to overlapping numbers. Should revise these equations to ensure they are clearly readable and correctly formatted. Special attention should be given to the proper display of subscripts and superscripts.

-The discussion on results has to be deepened in order to better highlight the advantages provided by the proposed approach.

Reviewer 4 ·

Basic reporting

very poorly written, multiple grammatical, punctuation mistakes , including long sentences in abstract and other sections without connectivity (such as the first starting sentence of abstract)

the author claims to bring efficiency without computation with many other parameters however this has not been justified by comparing to multiple state of the art work.

Author has shown some recent achievements of [16, 17] but has not compared his work with it. and has only relied on one comparison of previous work.

Experimental design

Figure 2 has to be revised , not visible
explain the algorithm in text a little more, not enough
Figure 8 Packets Drop Comparison Line number 449, looks awkward , the details explanation is missing ?

Validity of the findings

looks good but why the comparison has been stick to only one recent work that is EERSDRA?

Additional comments

Figurer and Tables are to be revised, with proper heading, labels and units used,
formatting issues to be removed,

---

## Round 0.2 · Minor Revisions

Thank you for submitting your manuscript to PeerJ Computer Science. We have now received the reviewers’ feedback and completed our evaluation of your submission.

Overall, the reviewers found your work to be of merit, but they have raised a few minor issues that need to be addressed before the manuscript can be accepted. Specifically, they highlighted the presence of some grammatical errors in the English language and the inconsistent use or incorrect declaration of acronyms throughout the paper.

We kindly request that you revise the manuscript to correct these issues. In particular, please ensure that all acronyms are clearly defined upon first use and that the English language is carefully reviewed to improve clarity and readability.

We believe these revisions will strengthen your manuscript and look forward to receiving your updated submission.

Reviewer 1 ·

Basic reporting

no comment

Experimental design

no comment

Validity of the findings

no comment

Additional comments

The authors have accommodated changes recommended by reviewer.

Reviewer 2 ·

Basic reporting

Authors have improved the quality of the paper, however, some minor flaws are observed in the reporting section.

Some abbreviations are still used without defining their full forms, (ref. abstract section).

Experimental design

Authors have not incorporated below comment in Table 3

Authors have mentioned only few simulation parameters (Table 3), some important parameters are missing, for example, simulation tool, runtime, rounds, MAC protocol,

Validity of the findings

Abstract and conclusion section must contain numerical findings and comparison with existing works

Additional comments

None.

Reviewer 3 ·

Basic reporting

I am satisfied with the latest revision

Experimental design

no comment

Validity of the findings

no comment

---

## Round 0.3 · accepted · Accept

I hope this message finds you well. After carefully reviewing the revisions you have made in response to the reviewers' comments, I am pleased to inform you that your manuscript has been accepted for publication in PeerJ Computer Science.

Your efforts to address the reviewers’ suggestions have significantly improved the quality and clarity of the manuscript. The changes you implemented have successfully resolved the concerns raised, and the content now meets the high standards of the journal.

Thank you for your commitment to enhancing the paper. I look forward to seeing the final published version.

Reviewer 1 ·

Basic reporting

no comment

Experimental design

no comment

Validity of the findings

no comment

Reviewer 2 ·

Basic reporting

No comment. All is okay!

Experimental design

No comment. All is okay!

Validity of the findings

No comment. All is okay!

Additional comments

I think the authors have incorporated all concerns and feedback, So I am happy to accept the article. Best of luck.

Reviewer 3 ·

Basic reporting

no comment

Experimental design

no comment

Validity of the findings

no comment